# Ethylenediaminetetraacetic Acid Disodium Salt Acts as an Antifungal Candidate Molecule against *Fusarium*
*graminearum* by Inhibiting DON Biosynthesis and Chitin Synthase Activity

**DOI:** 10.3390/toxins13010017

**Published:** 2020-12-27

**Authors:** Xiu-Shi Song, Kai-Xin Gu, Jing Gao, Jian-Xin Wang, Shao-Chen Ding, Mingguo Zhou

**Affiliations:** 1Key Laboratory of Pesticides, College of Plant Protection, Nanjing Agricultural University, Nanjing 210095, China; songxs@njau.edu.cn (X.-S.S.); xinkaigu@126.com (K.-X.G.); gjing5898@163.com (J.G.); jianxin-wang@njau.edu.cn (J.-X.W.); shaochend@126.com (S.-C.D.); 2The Key Laboratory of Plant Immunity, Nanjing Agricultural University, Nanjing 210095, China

**Keywords:** *Fusarium*, EDTANa_2_, deoxynivalenol, chitin synthases, manganese ion

## Abstract

*Fusarium* fungi are the cause of an array of devastating diseases affecting yield losses and accumulating mycotoxins. Fungicides can be exploited against *Fusarium* and deoxynivalenol (DON) production. However, *Fusarium* resistance to common chemicals has become a therapeutic challenge worldwide, which indicates that new control agents carrying different mechanisms of action are desperately needed. Here, we found that a nonantibiotic drug, ethylenediaminetetraacetic acid disodium salt (EDTANa_2_), exhibited various antifungal activities against *Fusarium* species and DON biosynthesis. The infection of wheat seeding caused by *F. graminearum* was suppressed over 90% at 4 mM EDTANa_2_. A similar control effect was observed in field tests. Mycotoxin production assays showed DON production was significantly inhibited, 47% lower than the control, by 0.4 mM EDTANa_2_. In vitro experiments revealed a timely inhibition of H_2_O_2_ production as quickly as 4 h after amending cultures with EDTANa_2_ and the expression of several *TRI* genes significantly decreased. Chitin synthases of *Fusarium* were Mn^2+^-containing enzymes that were strongly inhibited by Mn^2+^ deficiency. EDTANa_2_ inhibited chitin synthesis and destroyed the cell wall and cytomembrane integrity of *Fusarium*, mainly via the chelation of Mn^2+^ by EDTANa_2_, and thus led to Mn deficiency in *Fusarium* cells. Taken together, these findings uncover the potential of EDTANa_2_ as a fungicide candidate to manage Fusarium head blight (FHB) and DON in agricultural production.

## 1. Introduction

*Fusarium* is a globally important genus of fungal pathogens, responsible for many devastating diseases of plants and various serious diseases of humans [1,2]. *Fusarium* species are widely present in soil, plants and other organic substrates and have widespread distributions [3]. Species such as *Fusarium graminearum*, *Fusarium oxysporum* and *Fusarium verticillioides* can infect many crop plants, vegetables and flowers [2,4,5,6,7,8]. One of the major diseases caused by *Fusarium* is Fusarium head blight (FHB), which is becoming more and more serious recently and causing concern. FHB results in yield loss and damaging of cereal grains [9,10]. Additionally, *Fusarium* spp. produce various types of mycotoxins, including deoxynivalenol (DON) and acetyl-deoxy-nivalenol (3-ADON and 15-ADON), that suppress humoral and cellular immunity and are thus highly detrimental to human and animal health [11,12]. DON is a mycotoxin virulence factor that promotes growth of the *F. graminearum* fungus in wheat floral tissues [13].

Practices used to control FHB and DON include rotation with nonhost crops and tillage [14], planting of resistant cultivars [15], and application of fungicides [16]. Essentially, current protective measures against *Fusarium* species mainly rely on fungicides, such as benzimidazole, triazole, demethylation inhibitor and quinone outside inhibitor. However, some of these can lead to the enhancement of DON biosynthesis in the infected wheat [17,18]. Therefore, new molecules are needed to control FHB and inhibit DON biosynthesis.

In addition, *Fusarium* isolates are susceptible to mutations that lead to phenotypes of tolerance towards common antifungal drugs [19,20]. Resistance to fungicides allows pathogens to survive fungicide treatment. The time taken for a new resistant mutant to reach a population size that is unlikely to die out by chance is called “emergence time”. Prolonging emergence time would delay loss of control [21]. To date, the drawbacks of both scientific control strategies and the use of effective fungicides need to be addressed [8,22,23]. Undoubtedly, the processes of discovery and development of new antifungal drugs in the pharmaceutical industry are not only laborious and time consuming but also costly [24]. This is particularly true, as fungicides with novel modes of action are only rarely found, and resistance to single-target fungicides may occur within few years [25]. Some nonantibiotic drugs were recently reported to exhibit some antimicrobial activity against bacteria and *Candida albicans* [26,27].

Ethylenediaminetetraacetic acid (EDTA) is a chelating agent targeting divalent cations and has been previously used in oil fields to increase oil production and inhibit scale formation [28,29]. EDTA has also been shown to possess antimicrobial activities against bacteria and *C. albicans* because it can limit the availability of essential cations. The chelation of cations causes a separation of lipopolysaccharides from the outer membrane of microbial cells and thus increases the membrane permeability and subsequently leads to cell death in bacteria [30,31,32,33,34,35]. Generally considered safe, EDTA has been used intensively in the food and therapeutic industry [33,36,37]. Here, we demonstrated that EDTANa_2_ had antifungal activity against *Fusarium graminearum* and DON biosynthesis. Our study contributes to the understanding of the mechanisms underlying EDTANa_2_ control of FHB and provides a fungicide candidate molecule against *Fusarium graminearum* and its mycotoxin biosynthesis.

## 2. Results

### 2.1. Ethylenediaminetetraacetic Acid Disodium Salt Exhibits Various Antifungal Activities against Fusarium Species

Mycelial growth was inhibited with 0.15 mM ethylenediaminetetraacetic acid or ethylenediaminetetraacetic acid sodium salt (Figure 1A). The effectiveness of ethylenediaminetetraacetic acid sodium salt was affected by its number of sodium ions. For all tested reagents, EDTANa_2_ had the best antifungal activity in terms of growth inhibition as well as cell swelling effects, closely followed by EDTANa_3_. The antifungal activity of EDTANa_4_ was similar to that of EDTA. Moreover, the mycelial growth and morphology with the 0.3, 0.45, 0.6 mM sodium ion treatments showed no difference from the control group (only the treatment with 0.6 mM sodium ion is shown in Figure 1A), indicating that the sodium ions had no effect on *Fusarium* growth at a concentration less than 0.6 mM.

To test the antifungal activity of EDTANa_2_, thirteen isolates of *Fusarium*, including *F. acuminatum*, *F. asiaticum*, *F. avenaceum*, *F. concentricum*, *F. culmorum*, *F. equiseti*, *F. fujikuroi*, *F. graminearum*, *F. lateritium*, *F. oxysporum*, *F. proliferatum*, *F. solani* and *F. verticillioides* were inoculated into 96-well microtiter plates. The minimum inhibitory concentrations (MICs) of EDTANa_2_ on the test pathogens varied from 9.37 to 18.75 mM (Appendix A). The lowest MIC value was observed for *F. asiaticum*, *F. avenaceum*, *F. equiseti*, *F. fujikuroi* and *F. proliferatum*, while the other *Fusarium* isolates (except *F. lateritium*) had MIC values of 18.75 mM. The MIC value of EDTANa_2_ to *F. lateritium* was 200 mM. Surprisingly, EDTANa_2_ did not show any antifungal activity against *F. lateritium* but promoted its growth in the range of 4.69–150 mM.

For *F. graminearum* PH-1 strain, a linear regression of the percentage inhibition related to the control of mycelial growth versus the log_10_ transformation for each of EDTANa_2_ concentration was obtained. The median effective concentration (EC_50_) was calculated for each strain using a linear equation. The EC_50_ value of EDTANa_2_ for *F. graminearum* PH-1 strain was 0.29 mM (107.88 mg L^−1^) (Figure 1B).

### 2.2. The Control Effect and Phytotoxicity Test of Ethylenediaminetetraacetic Acid Disodium Salt

The EDTANa_2_ for control of seedling blight in wheat was effective, reducing disease severity by 59%, 79%, 92% at 1 mM, 2 mM and both 4 mM and 8 mM, relative to the inoculated control, respectively (Figure 2A). No significant different effect was observed among 1 mM to 8 mM EDTANa_2_, indicating that the control effect was stable within that range under controlled conditions. To ascertain whether the EDTANa_2_ molecule could inhibit *Fusarium* infection under natural conditions, a crop phytopathogen, *F. graminearum,* was chosen for pathogenicity assays by spray inoculation experiments. The disease incidences were recorded 21 days post inoculation (dpi). The field experiment was conducted for two years (2018 and 2019) and produced similar results to the previously described experiments. As shown in Figure 2B, EDTANa_2_ significantly reduced Fusarium head blight in the field. After spray treatment with 7 g ha^−1^ EDTANa_2_, *F. graminearum* caused 52% and 45% spikelets infection at 21 dpi, 45% and 49% reduction compared to the sterile water control in 2018 and 2019, respectively. When the dosage of EDTANa_2_ increased to 70 g ha^−1^, the incidence of disease decreased to 12% and 8%, 87% and 91% reduction compared to the sterile water control in 2018 and 2019, respectively. In 2019, the incidence of disease treated with 70 g ha^−1^ EDTANa_2_ had a 70% reduction compared to the 140 g ha^−1^ carbendazim treatment (Figure 2B). To further analyze the influence of EDTANa_2_ treatments on spikelet morphology, we sprayed a series of EDTANa_2_ concentrations (7 g ha^−1^ to 4000 g ha^−1^) onto wheat spikelets, and the results showed that there was little change in spikelets morphology following spray application of 7 g ha^−1^ to 1600 g ha^−1^ EDTANa_2._ However, when the dosage of EDTANa_2_ increased to 2000 g ha^−1^, phytotoxicity was observed (Figure 2C). These results suggested that EDTANa_2_ could be used as a safe antifungal agent at low concentration.

### 2.3. EDTANa_2_ Decreases DON Biosynthesis and TRI Gene Expression of Fusarium Graminearum In Vitro

Because the mycotoxin DON is a virulence factor, we investigated the mycotoxin biosynthesis potential of strains under EDTANa_2_ treatment. To verify the ability of the EDTANa_2_ to limit toxin production, DON amounts were measured using a competitive ELISA-approach. As shown in Figure 3A, the DON production in TBI media was significantly inhibited by 0.4 mM and 0.8 mM EDTANa_2_, about 47% and 57.3% lower than the control group, respectively.

As several lines of evidence in the literature corroborate an important role for H_2_O_2_ in induction of toxin production, the accumulation of H_2_O_2_ upon EDTANa_2_ application was monitored using an in vitro assay. It showed that adding 0.4 mM and 0.8 mM EDTANa_2_ resulted in a decreased H_2_O_2_ content in the medium compared to the control as fast as 4 h after the start of the assay (Figure 3B). This indicated that EDTANa_2_ decreased the intracellular oxygen content within a short time after adding to the medium, which may reduce the activation of oxygen to toxin synthesis.

To further reveal the expression profiles of individual genes and their coordination in the trichothecene biosynthesis pathway, we measured the expression of several *TRI* genes after treating with EDTANa_2_. We found four genes in the trichothecene biosynthesis pathway were significantly down-regulated compared with the control group (Figure 3C). After treating with 0.4 mM EDTANa_2_ at 12 h, the expression of the *TRI6* and *TRI10* genes, which have been identified as positive transcription factor genes for trichothecene biosynthesis in *F. graminearum*, decreased by 2- and 2.13-times; The *TRI12* gene, which is associated with trichothecene accumulation and resistance in *F. graminearum*, decreased by 3.45-times compared with the control group; The expression of the *TRI11* gene also decreased by 1.56-times. The inhibitory effect of 0.8 mM EDTANa_2_ on the *TRI* genes expression was consistent with that of 0.4 mM EDTANa_2_. It should be mentioned that the *Tri101* gene encoding a trichothecene 3-O-acetyltransferase showed a remarkable up-regulation, with 2.24- and 3.02-fold more transcripts after treating with 0.4 mM and 0.8 mM EDTANa_2_, respectively (Figure 3C). These in vitro results suggested that EDTANa_2_ can indeed inhibit DON biosynthesis and may be useful for reducing DON contamination in grains caused by *F. graminearum*.

### 2.4. Ethylenediaminetetraacetic Acid Disodium Salt Affected Cell Wall Formation and Cell Permeability

The inhibition of *Fusarium* growth by EDTANa_2_ was observed, so we analyzed the cell wall formation and cell permeability of cells grown in the presence of EDTANa_2_. The mycelia treated by EDTANa_2_ were sensitive to the preparation condition of scanning electron microscopy (SEM), and were destroyed and lysed with SEM observation (Figure 4A). As shown in Figure 4B, the ultra-structures of the untreated cells exhibited normal electron-dense layers and patterns under transmission electron microscopy. In contrast, the cells cultured in 0.075 mM EDTANa_2_ had a clearly altered cell wall electron density, thickness and ultrastructure. The most notable visual alteration was a thicker cell wall in the treated cells, which topped 573 nm compared with that of the control cell, which had an average of 142 nm (the biggest thickness was 153 nm) (*n* = 20). Furthermore, the number of layers in the treated cell wall was reduced, and the electron density declined, whereas three layers were easily recognized in the normal cell wall. In filamentous fungi, chitin, a β-1,4-linked polysaccharide of N-acetylglucosamine, is a key structural component of the cell wall [38]. To further analyze the effect of EDTA and EDTANa_2_ on cell wall formation, we measured the chitin content of *F. graminearum* cell walls after EDTA or EDTANa_2_ treatment. The results illustrated that the EDTA and EDTANa_2_ treatment groups produced only 33.43% and 25.23% chitin relative to the hyphal dry weight, values 35.35% and 51.2% lower than those of the control group, respectively (Figure 4C). It is noteworthy that EDTANa_2_ was more effective than EDTA, which concurred with our above results.

In most cases, the mode of action of antimicrobials against pathogens depends on the destruction of the fungal cell membrane and the resulting increase in cell permeability. The change in electrical conductivity reflects the change in the cell membrane permeability of *Fusarium*. Additionally, our data showed that the relative conductivity of hypha was significantly increased by 11% and 15% after 0.15 mM EDTA and 0.15 mM EDTANa_2_ treatment compared with that of the control group, respectively (Figure 4D). There was no difference between 0.6 mM NaCl treatment and the control group. Therefore, EDTA and EDTANa_2_, especially EDTANa_2_, indeed affected both cell wall formation and cell permeability and resulted in a decreasing resistance to the external environment in *F. graminearum*.

### 2.5. The Inhibitory Effects of Ethylenediaminetetraacetic Acid Disodium Salt against Fusarium are Ameliorated by Mn^2+^ but not Mg^2+^ and Ca^2+^

In order to find out the mechanism of inhibiting effect of EDTANa_2_, we examined the effect of the saturation of EDTANa_2_ (0.15 mM) by the addition of excess cations (0.15 mM to 1.2 mM). As shown in Figure 5A, the adding of Mn^2+^ increased the biomass of mycelia, but the adding of Mg^2+^ and Ca^2+^ had no significant effect on mycelia biomass. In addition, when an additional 0.3 mM MgCl_2_, 0.3 mM CaCl_2_ or 0.15 mM FeCl_3_ was added into the media as sources of divalent ions to bind 0.15 mM EDTANa_2_, mycelial growth was not resumed. 0.15 mM EDTANa_2_ plus 0.3 mM Mn^2+^, however, rescued mycelial growth and morphology, which was similar to that in control wells not given EDTANa_2_ (Figure 5B). That indicated that the inhibiting effect of EDTANa_2_ was related to its function of chelation.

We further measured the trace element content of *Fusarium* hyphae in different treatments. As shown in Figure 5C, when EDTA and EDTANa_2_ were added to the medium, the Mg and Mn elements decreased as expected. However, the contents of Ca and Fe elements were not significantly changed after the addition of EDTA and EDTANa_2_. When an additional Mg^2+^ or Ca^2+^ was added into the medium containing EDTANa_2_, all the trace elements, except Mn, increased or showed no difference compared with the untreated group. The Mn was always in a deficient state in the mycelia treated by EDTA and EDTANa_2_ unless Mn^2+^ was added into the medium. These results suggested that intracellular Mn^2+^ was chelated by EDTANa_2_, resulting in cell wall and cell membrane defects.

### 2.6. Chitin Synthases are More Active in F. graminearum When Mn^2+^ Is Used as a Cofactor

To further study the mechanism of the EDTANa_2_ control effect, chitin synthases were extracted for enzymatic analysis. Unlike most chitin synthases in other fungi that used Mg^2+^ as the cofactor, chitin synthases in *F. graminearum* were more active when Mn^2+^ was used at the active site. The chitin synthases with 5 mM Mn^2+^ had a 26% higher activity than that with 5 mM Mg^2+^, indicating that chitin synthases in *F. graminearum* may use Mn^2+^ as the cofactor. The addition of 0.25 mM EDTANa_2_ into the reaction systems could decrease enzymatic activity by 40% and 44% after chelation of Mn^2+^ and Mg^2+^, respectively. However, chitin synthase in the Mn^2+^ reaction system still had a 33% higher activity than that in the Mg^2+^ reaction system. When the EDTANa_2_ concentration was increased to 1 mM, the activity of chitin synthase in the Mn^2+^ and Mg^2+^ reaction systems decreased to 0.09 and 0.11 nmol GlcNAc h^−1^ mg^−1^, respectively (Figure 6A). Consistent with this, the metal ion addition experiment showed that the chitin content returned to normal levels after the addition of 0.15 mM Mn^2+^ to SNA culture medium (containing 0.15 mM EDTANa_2_). While wells adding Mg^2+^ or Ca^2+^ produced equivalent chitin to wells adding EDTANa_2_ only, a 35–50% reduction was observed compared to the control group (Figure 6B).

## 3. Discussion

The genus *Fusarium,* which is pathogenic and toxic to plants and humans, is one of the most economically damaging fungal genera. *Fusarium* colonizes a wide range of environments, and control of *Fusarium* floral infections remains problematic. Some fungicides, including triazoles, benzimidazoles and strobilurins, are moderately effective, but reports of fungicide failure resulting from resistance have increased since the late 1960s [39,40]. To make matters worse, the development of new fungicides is much slower than the appearance of fungicide resistance in *Fusarium* populations [41,42]. Here, a nonantibiotic drug, EDTANa_2_, exhibited novel antifungal activity against *F. graminearum* and DON production. Through combined analyses of morphology, DON content and *TRI* gene expression detection, chitin detection, cell permeability, transmission electron microscopy and field experiments, we demonstrated that EDTANa_2_ destroyed cell wall and cytomembrane integrity and inhibited *TRI* gene expression in *F. graminearum*, and the antifungal effect of EDTANa_2_ relied on Mn^2+^-chelating abilities.

EDTA is considered as an ion chelator, and it has been recommended as an antimicrobial agent against bacteria and *C. albicans* [43,44,45]. The EC_50_ value of EDTANa_2_ for *F. graminearum* is 107.88 mg L^−1^, which is higher than that of carbendazim (about 0.5 mg L^−1^). The field test, however, shows that 70 g ha^−1^ EDTANa_2_ decreased the incidence of disease by 70% compared to 140 g ha^−1^ carbendazim treatment. As DON is an important virulence factor in wheat, previous studies suggest that significant decreases in FHB incidence in field situations are possible with proper DON inhibited fungicide applications [46]. In our study, the DON production was significantly inhibited by EDTANa_2_, which may increase the antifungal effects on FHB.

The fungal cell wall protects the cell against osmotic pressure and other environmental stresses and is considered the carbohydrate armor of the fungal cell [47]. After EDTANa_2_ treatment, all layers across the cell wall were affected, and the chitin content was decreased in *Fusarium*. In addition, the membrane permeability increased significantly when *Fusarium* was cultured in medium containing EDTANa_2_. This suggests that the fungal cells became more sensitive to changes in environmental stresses with EDTANa_2_ treatment.

Our next question was which divalent cation was chelated by EDTANa_2_ and then caused chitin content reduction and cell wall defects in *Fusarium*. There have been reports that calcium-binding agents inhibit *Cryptococcus neoformans* and *C. albicans* by disrupting the assembly of the polysaccharide capsule through Mg^2+^ and Ca^2+^ chelation [31,48,49]. However, the results in our paper showed that EDTANa_2_ chelated Mn^2+^ and resulted in a reduction of chitin synthesis. In fungi, chitin is synthesized by chitin synthase, whose activities are known to depend upon the presence of a divalent cation [50,51]. Chitin synthases (CHSs) in *Fusarium* were previously classified into seven categories [52]. Different chitin synthases are distinct in their responses to the divalent cation, for example, Chs2 and Chs3 are stimulated, while Chs1 is inhibited by Co^2+^ in *Saccharomyces cerevisiae* [53]. On the basis of the data obtained in our study, it may be logical to assume that Mn^2+^ is essential for the main chitin synthase of *Fusarium*. EDTA had a fungistatic effect on *F. fujikuroi* growth, a pathogen causing bakanae disease, and its action was largely suppressed by Mn^2+^ and slightly by Ca^2+^ [54]. Combining with our findings, we can speculate that this kind of chelating agent, such as EDTA and EDTANa_2_ inhibits *Fusarium* spp. mainly because of Mn^2+^ deficiency.

In summary, EDTANa_2_ inhibits DON production and disrupts the cell wall and cell membrane functionality of *Fusarium*, an effect that appears to mainly result from Mn^2+^ chelation. The results of our study provided new material and candidate compound against *Fusarium* in crop protection.

## 4. Materials and Methods

### 4.1. Fungi, Plants, and Culture Conditions

*Fusarium graminearum* strains PH-1, *F. asiaticum* strain 2021, *F. acuminatum*, *F. avenaceum*, *F. concentricum*, *F. culmorum*, *F. equiseti*, *F. fujikuroi*, *F. lateritium*, *F. oxysporum*, *F. proliferatum*, *F. solani* and *F. verticillioides* (Appendix A) used in this study were stored in our laboratory [55]. The wheat variety Huaimai33 was maintained in our laboratory. Carboxymethyl cellulose (CMC) broth [56] and SNA medium (0.1% KH_2_PO_4_, 0.1% KNO_3_, 0.05% MgSO_4˙_7H_2_O, 0.05% KCl, 0.02% glucose, and 0.02% sucrose) were used for conidia production and assessments of mycelial growth, respectively. EDTA (99%) and inorganic metal salts were purchased from Sigma-Aldrich (St. Louis, MO, USA).

### 4.2. Control Effect Measurement on Wheat Seedling Blight

The test was evaluated under controlled conditions using a completely randomized design with two replications for each treatment. Ten seedlings per treatment were inoculated on the fully expanded primary leaves 8 d after Huaimai33 planting. The EDTANa_2_ was sprayed on leaves at different concentration from 0.5 to 8 mM. After 24 h, leaves were punctured and inoculated with three microliter of macroconidia suspension (1 × 10^6^ spores mL^−1^). The lesions of diseased leaves were measured and photographed at 6th day post inoculation. Duncan’s multiple comparison test (SPSS20.0, IBM, Chicago, IL, USA) with a significant difference set as *P* < 0.05 was used to compare sample means. Mean values and standard deviations were reported. The differences between means with *P* less than 0.05 were considered statistically significant. The control effect was characterized by linear regression analysis (R = 0.94) using the SPSS statistical package.

### 4.3. Control Effect Measurement on Fusarium Head Blight and Phytotoxicity Field Test

The field study was conducted for two years (2018 and 2019) at the same location with different randomizations for each year (Table 1). Wheat (*Triticum aestivum* L. cv. Huaimai33) was grown on the experimental farm of Nanjing Agricultural University. At Zadok’s growth stage (ZGS) 65, while more than half of the wheat spikes were in bloom, field plots were arranged in a randomized block, which was designed with three 3 plots (each plot was 4 × 5 m). The treatments were as follows: (1) a control consisting of water; (2) 140 g ha^−1^ carbendazim; (3) 7 g ha^−1^ EDTANa_2_; (4) 70 g ha^−1^ EDTANa_2_. 7–4000 g ha^−1^ EDTANa_2_ were used for phytotoxicity assays. For floral spray inoculations, each plant was sprayed with 0.5 mL of 1 × 10^4^ spores mL^−1^
*F. graminearum* strain PH-1 conidia 24 h after EDTANa_2_ spray treatment. Pathogenicity assays were performed 14 or 21 days after EDTANa_2_ spray treatment as described previously [57]. The influence of EDTANa_2_ on wheat was tested by assessing browning spikelets. Thirty wheat heads were randomly selected to calculate the browning spikelets ratio for each concentration of EDTANa_2_. The browning spikelets ratio was defined as follows: browning spikelet (%) = browning spikelets/total spikelets. The experiment was replicated three times.

### 4.4. Mycelial Growth Inhibition by EDTANa_2_

For a fungicide-sensitivity assay in the laboratory, a three-day-old mycelial plug (5 mm in diameter) was placed in the center of a minimal medium (MM) (10 mM K_2_HPO_4_, 10 mM KH_2_PO_4_, 2.5 mM NaCl, 4 mM NH_4_NO_3_, 10 mM glucose) plate amended with EDTANa_2_ at 0, 0.1, 0.2, 0.4, 0.8, 1.6 mM (0, 37.2, 74.4, 148.8, 297.6, or 595.2 mg L^−1^). After 4 d at 25 °C, the colony diameters in two perpendicular directions of each plate were measured and averaged. Each combination of strain and concentration was represented by three biological replicates. The median effective concentration (EC_50_) value was calculated with DPS software (version 7.0, DPS Inc., Cary, NC, USA).

### 4.5. Optical, Scanning Electron Microscopy and Transmission Electron Microscopy Observation

Morphological observation of mycelia was performed using an inverted Olympus IX71 microscope (Olympus Canada, Markham, ON, Canada). Images were captured and analyzed by Image-Pro Plus 4.5 software (Media Cybernetics, Silver Spring, Maryland). Scanning electron microscopy (SEM) and transmission electron microscopy (TEM) were carried out using hyphae germinated from spores in SNA supplemented with 0.15 mM EDTANa_2_ at 25 °C for 24 h. All the cultures were performed in triplicate. For SEM determination, mycelia were fixed with 2.5% glutaraldehyde in 0.1 M sodium phosphate buffer (pH 7.2) at 4 °C for 12 h. The samples were then washed with sodium phosphate buffer (0.1 M, pH 7.2) and treated with 1% osmium tetroxide in sodium phosphate buffer for 1 h, subjected to gradual dehydration in ethanol (70, 80, 90 and 100%), and dried to the critical point (CPD 030 Critical Point BALTEC Dryer, Leica Microsystems, Liechtenstein). After drying, the samples were glued on stubs using carbon tape and coated with gold (Sputter Coater BALTEC SDC 050, Leica Microsystems, Liechtenstein). For TEM determination, sections were prepared and visualized using a H-7650 transmission electron microscope (Hitachi, Tokyo, Japan) as described by Song et al. [57].

### 4.6. Measurement of H_2_O_2_ and DON

H_2_O_2_ formation in the fungicide experiments was measured 4 h and 12 h after 0.4 mM or 0.8 mM EDTANa_2_ treatment using a TMB (tri-methyl-benzidin) assay [58]. H_2_O_2_ formation was determined by measuring the absorbance at 620 nm in duplicate in each time point and in three independent experiments. In each experiment, a standard curve of pure H_2_O_2_ was added in a concentration range of 0.01 mM up to 100 mM. The H_2_O_2_ formed in the in vitro assay was calculated based on this standard curve.

DON production in TBI cultures was assayed with a competitive ELISA-based DON detection plate kit (Wise, Zhenjiang, China) according to previous studies [59]. Ten microliter of conidia (1.5 × 10^7^/mL) were inoculated in 30 mL TBI and cultured at 28 °C for 24 h in dark, and then 0.4 mM or 0.8 mM EDTANa_2_ was added and cultured for additional 6 days. The experiment was repeated three times. To assay *TRI* gene expression, hyphae were harvested from 2-day-old TBI cultures (1 day after EDTANa_2_ adding) and used for RNA isolation. qPCR was performed as previously describe (Appendix A) [57]. The tubulin gene of *F. graminearum* was used as the internal control. The results were calculated with the data from three biological replicates.

### 4.7. Chitin Content, Chitin Synthase Activity and Cell Membrane Permeability Measurement

Macroconidia (10^4^ mL^−1^) were cultured in SNA, SNA amended with 0.15 mM EDTA or SNA amended with 0.15 mM EDTANa_2_ for 7 days and were used for chitin determination as previously described [57]. Fresh mycelium cultured in Czapek’s medium (3 g L^−^^1^ of NaNO_3_, 1.31 g L^−^^1^ of K_2_HPO_4_, 0.5 g L^−^^1^ of KCl, 0.5 g L^−^^1^ of MgSO_4_·7H_2_O, 0.01 g L^−^^1^ of FeSO4·7H_2_O, 30 g L^−^^1^ of sucrose, pH 7.2) for 5 d were collected and finely ground with liquid nitrogen for chitin synthase extraction. Chitin synthase activity was extracted and measured according to Song et al. [57] with some modifications. Chitin synthase activity was measured by following (^14^C) GlcNAc incorporation into the filter-retainable polymer in the presence of 0–1 mM EDTANa_2_ plus 5 mM Mg^2+^ or Mn^2+^. To measure the cell membrane permeability, macroconidia (final concentration was 10^3^ mL^−1^) were inoculated into SNA, SNA with 0.3 mM NaCl, SNA with 0.15 mM EDTA or SNA with 0.15 mM EDTANa_2_ at 25 °C for 7 days. The conductivity was measured with a conductometer (CON510 Eutech/Oakton, Singapore) as described previously [60]. The conductivity of mycelia boiled for 5 min represented the final conductivity. The relative conductivity was calculated as follows: relative conductivity (%) = conductivity/final conductivity × 100. Three biological replicates were tested for each treatment.

### 4.8. Fungicidal Activity of EDTANa_(n)_ against Fusarium spp.

Ten microliters of spores (10^4^ mL^−1^) of *F. graminearum* PH-1 were plated in a 96-well flat bottom culture plate containing 0.15 mM EDTA, EDTANa_2_, EDTANa_3_ or EDTANa_4_ in 100 μL of SNA medium (pH 4.2). The plates were cultured at 25 °C for 24 h and observed using an inverted microscope. The growth and morphology of mycelia were photographed and compared among treatments using NIS-Elements AR software (version 3.2, Nikon, Tokyo, Japan). The experiment was replicated three times.

The determination of the minimum inhibitory concentration (MIC) was performed using 96-well microtiter plates. The *Fusarium spp.* (listed in ‘Fungi, plants, and culture conditions’) fungal inoculated in 96-well microtiter plates were treated with EDTANa_2_ at different concentrations and incubated for 24 or 36 h. The lowest concentration that demonstrated no visible growth was determined as the MIC. Measurements were repeated three times.

### 4.9. Effect of Metal Ions on EDTANa_2_ Activity

To assess whether cations would ameliorate the inhibitory effects of EDTANa_2_ on mycelial growth and chitin synthesis, an additional 0.3 mM MgCl_2_, 0.3 mM CaCl_2_, 0.3 mM MnCl_2_ or 0.15 mM FeCl_3_ was added into SNA medium separately to bind the preadded 0.15 mM EDTANa_2_, followed by inoculation at 25 °C for 24 h and subsequent micro-examination. Corresponding amounts of MgCl_2_, CaCl_2_, MnCl_2_ or FeCl_3_ were used as controls. Subsequently, a series of MgCl_2_, CaCl_2_ or MnCl_2_ concentrations (0, 0.15, 0.3, 0.45, 0.6, 1.2 mM) was added to the media to saturate the 0.15 mM EDTANa_2_ to different degrees, and the fungi were cultured for 52 h to measure the mycelial biomass at OD_290_. The mycelia that were treated with 0.3 mM cations were collected at 7 d, frozen and dried for chitin content and trace element measurements [61]. Three independent experiments were performed, and the average was calculated.

### 4.10. Statistical Analysis

Statistical analysis was performed using Duncan’s multiple comparison test (for multiple comparisons) and Student’s *t*-test, all at a significance level of 0.05.

## Figures and Tables

**Figure 1 toxins-13-00017-f001:**
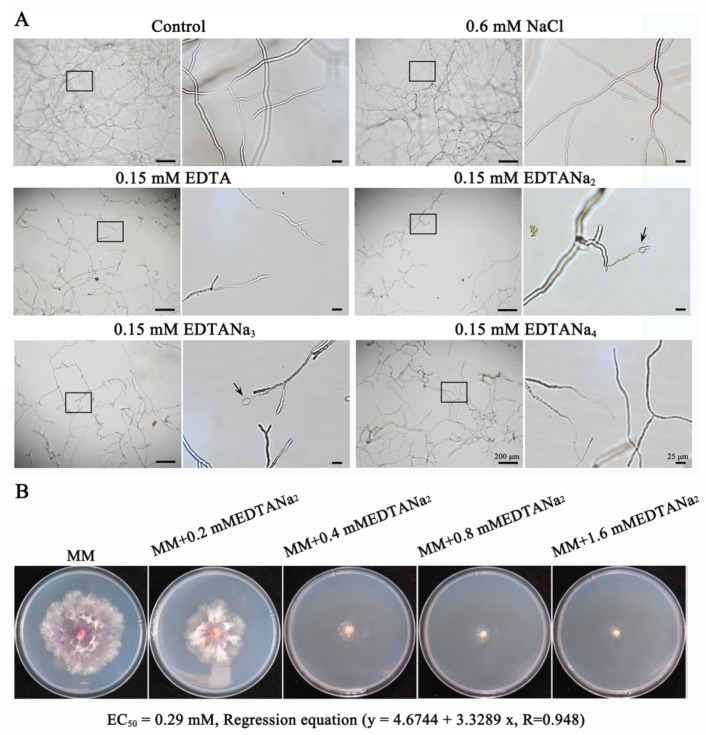
The antifungal activity of ethylenediaminetetraacetic acid disodium salt (EDTANa)_(*n*)_ against *Fusarium graminearum.* (**A**) The influence of sodium ion on EDTA effects. Conidia of *F. graminearum* were grown in SNA (1 g L^−1^ KH_2_PO_4_, 1.0 g L^−1^ KNO_3_, 0.5 g L^−1^ MgSO_4__˙_7H_2_O, 0.5 g L^−1^ KC1, 0.2 g L^−1^ glucose, 0.2 g L^−1^ sucrose) medium with 0.15 mM EDTANa_(*n*)_ (*n* = 0, 2, 3 or 4; pH 4.2) and were photographed after 24 h of cultivation. The effect of sodium ions was assayed by replacing EDTANa_(*n*)_ with NaCl and was used as the second control group. Each right panel is an enlarged view of the area in the black box in the left panel. The arrows indicate conglobate structures after 24 h in response to EDTANa_(*n*)._ The experiment was repeated three times with the same patterns. (**B**) Sensitivities of *F. graminearum* to EDTANa_2_. The measurement was performed on minimal medium (MM) after 4 d at 25 °C. Photos were taken 3 days after incubation. The experiment was repeated three times with the same patterns.

**Figure 2 toxins-13-00017-f002:**
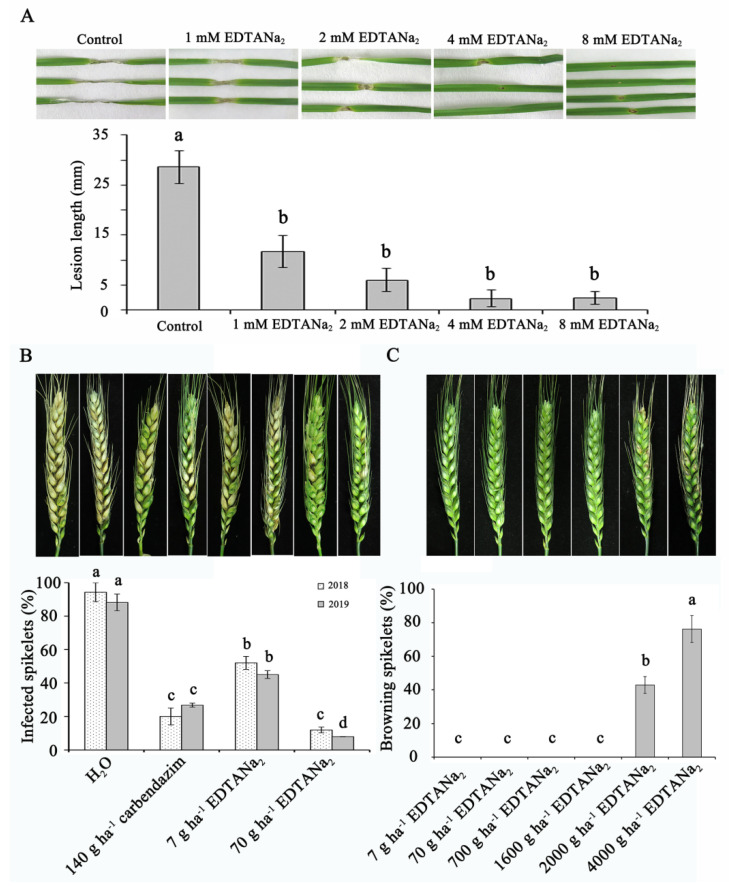
Control effect and phytotoxicity test of EDTANa_2._ (**A**) Control effect of EDTANa_2_ on wheat seedling blight in laboratory. The test was designed with two random replications for each race of plant. Ten seedlings per treatment were inoculated with macroconidia suspension 24 h after EDTANa_2_ spray. The lesions of diseased leaves were measured and photographed on sixth day post inoculation. Different letters represent a significant difference at *p* < 0.05. A linear regression equation of the percentage control effect for each EDTANa_2_ concentration was developed using the SPSS 20.0 (IBM, Chicago, IL, USA) statistical package (*y* = 4.67 + 3.329*x*, R = 0.9476). (**B**) Control effect of EDTANa_2_ on Fusarium head blight in the field. Wheat spikelets (cultivar Huaimai33) were sprayed with water, 40% carbendazim (140 g ha^−1^), or EDTANa_2_. Twenty-four hours later, the spikelets were inoculated via a spray inoculation experiment with a conidial suspension. Each combination of fungicide treatment and fungus was represented by 30 heads. After 21 days, the percentages of infected spikelets were determined, and representative heads were photographed. Values are means ± SD. Different letters represent a significant difference at *P* < 0.05. (**C**) Phytotoxicity test of EDTANa_2_ in the field. Wheat spikelets (cultivar Huaimai33) were sprayed with EDTANa_2_ and photographed at 21 days post inoculation. The data are an average ± standard error from 30 randomly selected heads. The experiment was replicated three times.

**Figure 3 toxins-13-00017-f003:**
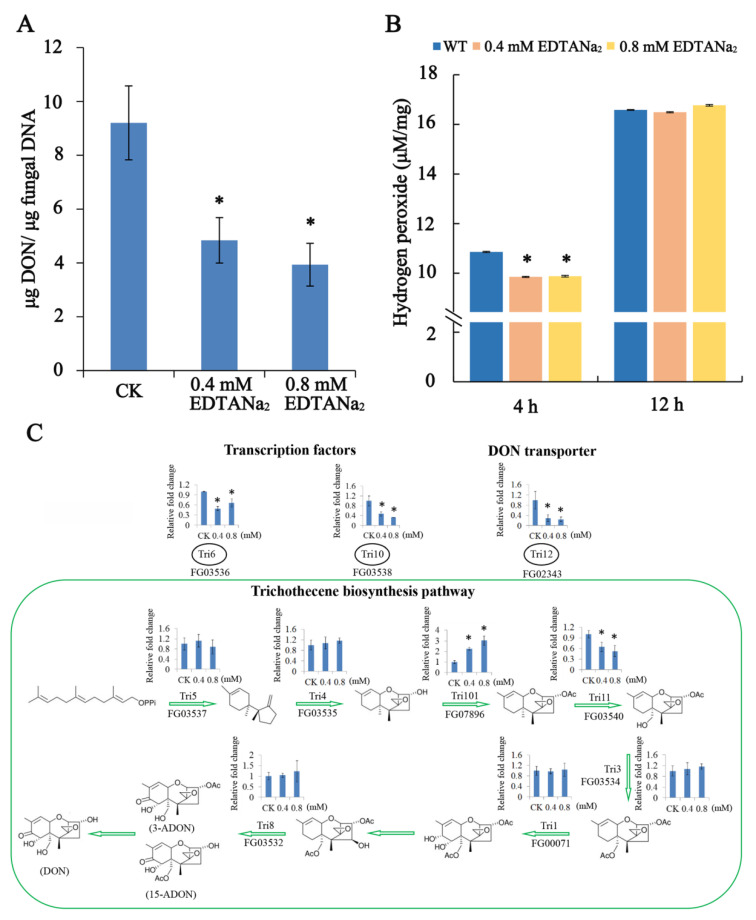
Effect of EDTANa_2_ on production of deoxynivalenol (DON) (**A**), H_2_O_2_ concentrations (**B**) and *TRI* genes expression (**C**). DON content was determined using a competitive ELISA approach 7 d after start of the experiments. The experiment was repeated three times. H_2_O_2_ was measured at 4 h and 12 h and calculated based on a standard curve included in each experiment. *TRI* gene expression was assayed by qRT-PCR. Hyphae were harvested from 2-day-old TBI cultures (1 day after EDTANa_2_ adding). Data are represented as the means ± SD of three biological replicates (significant differences at * *p* < 0.05).

**Figure 4 toxins-13-00017-f004:**
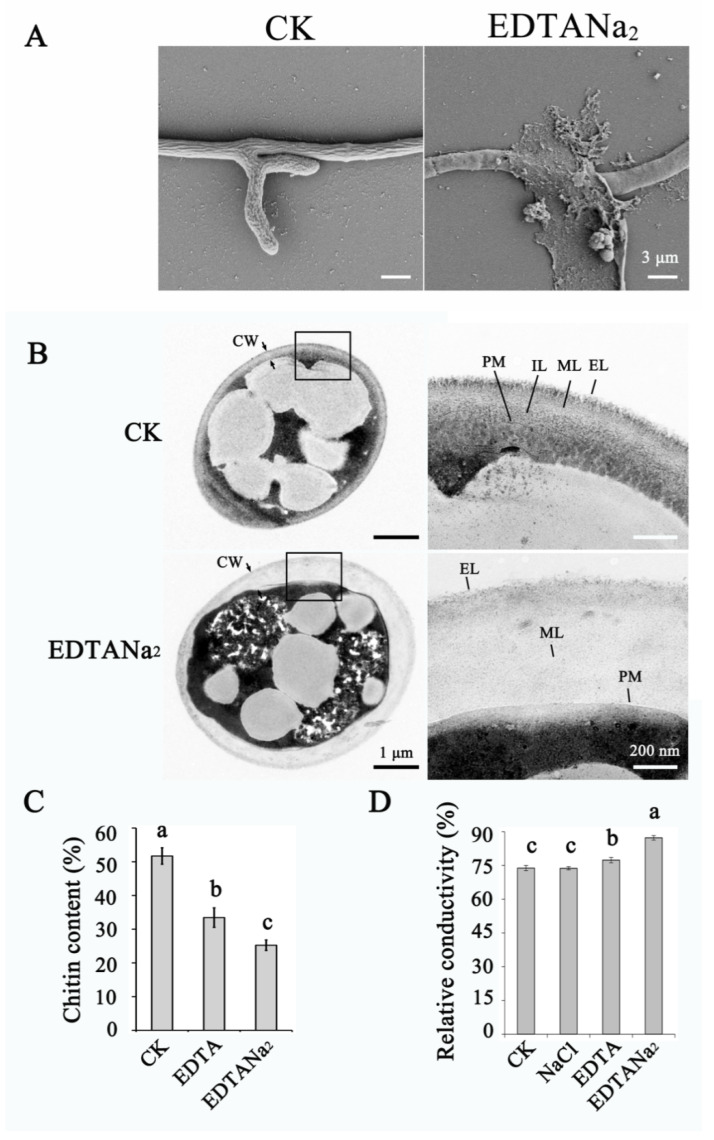
Effects of EDTANa_2_ on *Fusarium* cell wall formation and cell permeability. (**A**) Scanning electron microscopy of hyphae of *Fusarium graminearum* treated with or without EDTANa_2_. (**B**) Transmission electron microscopy of hyphae of *F. graminearum* treated with or without EDTANa_2_. The right panel is an enlarged view of the area in the black box in the left panel. CW, cell wall; EL, external electron-dense layer; ML, middle electron-dense layer; IL, internal electron-dense layer; PM, plasma membrane. Chitin content (**C**) and relative conductivity assay (**D**) of *F. graminearum* treated with 0.3 mM NaCl, 0.15 mM EDTA or EDTANa_2._ The experiment was performed in triplicate. Different letters represent a significant difference at *p* < 0.05.

**Figure 5 toxins-13-00017-f005:**
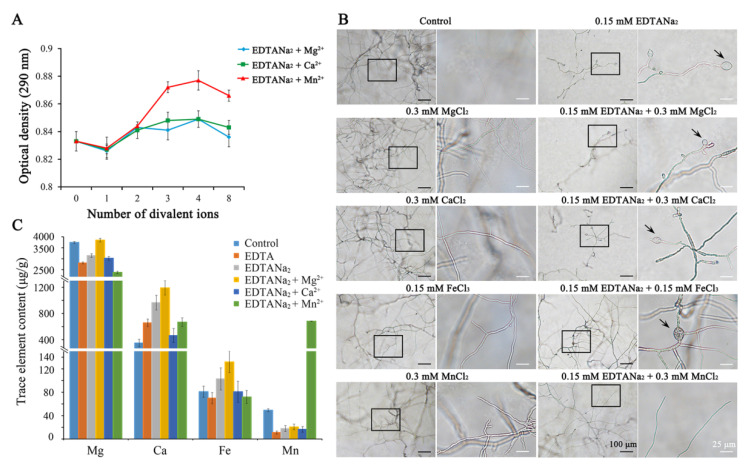
(**A**) EDTANa_2_ saturation with Mg^2+^, Ca^2+^ and Mn^2+^ induces the resumption of mycelial growth. The mycelial biomass was measured after 52 h of incubation as from MgCl_2_, CaCl_2_ or MnCl_2_ adding. Mycelial biomass was expressed as absorbance. (**B**) Effects of different cations on EDTANa_2_ activity against *Fusarium graminearum*. The right panel is an enlarged view of the area in the black box in the left panel. The arrows indicate conglobate structures. The same amounts of cations were added to the medium as controls. The experiment was repeated three times with the same patterns. (**C**) Concentration of selected trace elements in fungi treated with 0.15 mM EDTANa_2_ or 0.15 mM EDTANa_2_ plus 0.3 mM additional cations. Mycelial biomass and trace element assays were performed using three biological replicates for each group.

**Figure 6 toxins-13-00017-f006:**
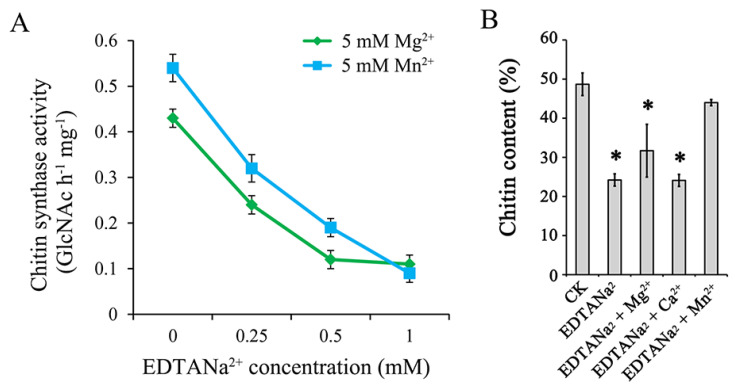
Chitin synthase activity and chitin content assay of *Fusarium graminearum*. (**A**) Chitin synthase activity was measured following (^14^C) GlcNAc incorporation into the filter-retainable polymer in the presence of Mg^2+^ or Mn^2+^. Data are expressed as nmol of GlcNAc incorporated per hour per mg of protein. (**B**) Mycelia were treated with EDTANa_2_ or EDTANa_2_ with additional cations. EDTANa_2_ (0.15 mM) was saturated by adding Mg^2+^, Ca^2+^ or Mn^2+^ (0.45 mM) to SNA media. Data are represented as the means ± SD of three biological replicates (* indicates significant differences at *p* < 0.05 compared to CK).

**Table 1 toxins-13-00017-t001:** Climate conditions in the field study.

Year	Experimental Date	Rainy Period	Temperature ^a^ (°C)		
			T_av_	T_max_	T_min_
2018	20 Apr.–4 May.	21 Apr.–23 Apr., 29 Apr.–1 May	21	30	11
2019	20 Apr.–11 May.	21 Apr., 27 Apr. and 28 Apr.	18	30	9

^a^ Data represent the average of the daily mean (T_av_), minimum (T_min_) and maximum (T_max_) temperatures recorded during each experimental period.

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
