# Peer review of "Ethylenediaminetetraacetic Acid Disodium Salt Acts as an Antifungal Candidate Molecule against Fusarium graminearum by Inhibiting DON Biosynthesis and Chitin Synthase Activity"

_toxins, 2020, doi:10.3390/toxins13010017_

Round 1

Reviewer 1 Report

The manuscript entitled “ Ethylenediaminetetraacetic acid disodium salt acts as 2 an antifungal candidate against Fusarium by 3 inhibiting DON biosynthesis and chitin synthase 4 activity” aims to find efficient nonantibiotic drug  which have antifungal activities agaginst Fusarium species and DON biosynthesis. This area of work could be intrested because biotic stress caused by Fusarium Head Blight has been presented in many parts of the world which can also influence wheat plants in terms of grain yield and quality.

In the manuscript there are only few misunderstandings.

Introduction

At the end of the introduction you are writing about results. That is not the place to be. It should be removed or moved to results.

Results

Pg 3. Lines 93-95, all Fusarium species should be writen in italic, as this is latin name

Pg 4. Lines 119-121 Those are to opposite/contraversal sentences. You need to correct this, for example in the last sentence you can write that EDTANa2 could be used as safe antifungal agent in small dosages….

Pg 4., Figure 2, in the graphs somewhere you have technical mistakes of not writing -1 in the exponent (ha-1)

Pg 9. Lines 215-216 you should not used references in the results

Pg 9, I am just thinking what would have hapened in the case that N, P or K would have been used, not Mg2+ or Ca2+ (as in the field conditions those elements are usually added as fertilizer)

In general, I think that results need to be more consistent, possibly shorter, so that someone who read article can understand this, without too much details.

Also, results seems somehow to short , compared to discussion.

Pg 12. As you recommend that EDTANa2 can be used as fungicide, I think that is to fast conclusion. Still you do not have so much field data to make such conclusion (also cost, compared to 'normal' fungicide it would be interesting to know)

Pg 12, lines 315-317 all Fusarium species should be writen in italic form

Reviewer 2 Report

Suggestions are in the enclosed pdf

Reviewer 3 Report

I have made several suggestions for spelling and grammar; please see the attached scanned manuscript.

Figure 2A is not described in the results section.

Wheat inflorescences are commonly called heads or ears.  Use either heads or ears, but be consistent throughout the manuscript.

The legend for figure 2A is very long.  Some of these details should be placed in the methods or results section.  I think MBC refers to carbendazim? 

I think the authors should comment on the transient nature of the hydrogen peroxide reduction (in response to EDTANa2 addition) in the results or discussion section.

Are there any data from the literature how EDTANa2 might be transported into the fungal cells?  Is it possible that EDTANa2 diffuses across the fungal membrane?  Answers to these questions could be added to the discussion if appropriate.

I don’t believe I found a description for the chitin synthase activity measurement in the materials and methods section.  The description in the figure 6 legend is not detailed enough.

Please amend section 4.8: line 406 would “fungal inocula” be spores or conidia?

Round 2

Reviewer 2 Report

No comments